# Genetic diversity and drug resistance pattern of *Mycobacterium tuberculosis* strains isolated from pulmonary tuberculosis patients in the Benishangul Gumuz region and its surroundings, Northwest Ethiopia

Tekle Airgecho Lobie[1,2,3¤]*, Yimtubezinash Woldeamanuel[2], Daniel Asrat[2], Demissew Beyene[1], Magnar Bjørås[3,4], Abraham Aseffa[1]

1 Armauer Hansen Research Institute (AHRI), Addis Ababa, Ethiopia, 2 Department of Microbiology, Immunology, and Parasitology (DMIP), Addis Ababa University, Addis Ababa, Ethiopia, 3 Department of Microbiology, Oslo University Hospital and University of Oslo, Oslo, Norway, 4 Department of Clinical and Molecular Medicine, Norwegian University of Science and Technology, Trondheim, Norway

¤ Current address: Department of Microbiology, Oslo University Hospital and University of Oslo, Oslo, Norway
* teklemicro@gmail.com

## Abstract

### Introduction

Tuberculosis (TB) remains a major global public health problem and is the leading cause of death from a single bacterium, *Mycobacterium tuberculosis* (MTB) complex. The emergence and spread of drug-resistant strains aggravate the problem, especially in tuberculosis high burden countries such as Ethiopia. The supposedly high initial cost of laboratory diagnosis coupled with scarce financial resources has limited collection of information about drug resistance patterns and circulating strains in peripheral and emerging regions of Ethiopia. Here, we investigated drug susceptibility and genetic diversity of mycobacterial isolates among pulmonary tuberculosis patients in the Benishangul Gumuz region and its surroundings in northwest Ethiopia.

### Methods and material

In a cross-sectional study, 107 consecutive sputum smear-positive pulmonary tuberculosis (PTB) patients diagnosed at two hospitals and seven health centers were enrolled between October 2013 and June 2014. Sputum samples were cultured at Armauer Hansen Research Institute (AHRI) TB laboratory, and drug susceptibility testing (DST) was performed against Isoniazid, Rifampicin, Ethambutol, and Streptomycin using the indirect proportion method. Isolates were characterized using polymerase chain reaction (PCR)based Region of Difference 9 (RD9) testing and spoligotyping. Statistical analysis was performed using Statistical Package for the Social Sciences (SPSS) for Windows version 24.0.

**Data Availability Statement:** All relevant data are within the manuscript and its Supporting Information files.

**Funding:** This work is fully supported by AHRI core budget. AHRI receives core financial support from Swedish International Development Cooperation Agency (SIDA), Norwegian Agency for Development Cooperation (NORAD), and from the Government of Ethiopia through Ministry of Health. AA holds a grant from Human Hereditary and Health in Africa (H3Africa) [H3A/18/003] implemented by the African Academy of Sciences (AAS) and the NEPAD Agency's Alliance for Accelerating Excellence in Science in Africa (AESA) in partnership with Wellcome and a grant from the National Institutes of Health (NIH) Fogarty International Center Global Infectious Diseases entitled "Ethiopia-Emory TB Research Training Program" (D43TW009127).

**Competing interests:** The authors have declared that no competing interests exist.

## Results

Of 107 acid-fast-bacilli (AFB) smear-positive sputum samples collected, 81.3% (87/107) were culture positive. A PCR based RD9 testing revealed that all the 87 isolates were *M. tuberculosis*. Of these isolates, 16.1% (14/87) resistance to one or more drugs was observed. Isoniazid monoresistance occurred in 6.9% (6/87). Multidrug resistance (MDR) was observed in two isolates (2.3%), one of which was resistant to all the four drugs tested. Spoligotyping revealed that the majority, 61.3% (46/75) of strains could be grouped into ten spoligotype patterns containing two to 11 isolates each while the remaining 38.7% (29/75) were unique. SIT289 (11 isolates) and SIT53 (nine isolates) constituted 43.5% (20/46) among clustered isolates while 29.3% (22/75) were "New" to the database. The dominant families were T, 37% (28/75), CAS, 16.0% (12/75), and H, 8% (6/75), adding up to 51.3% (46/75) of all isolates identified.

## Conclusion and recommendations

The current study indicates a moderate prevalence of MDR TB. However, the observed high monoresistance to Isoniazid, one of the two proxy drugs for MDR-TB, reveals the hidden potential threat fora sudden increase in MDR-TB if resistance to Rifampicin would increase. Clustered spoligotype patterns suggest ongoing active tuberculosis transmission in the area. The results underscore the need for enhanced monitoring of TB drug resistance and epidemiological studies in this and other peripheral regions of the country using robust molecular tools with high discriminatory power such as the Mycobacterial Interspersed Repetitive Units -Variable Number of Tandem Repeats (MIRU-VNTR) typing and whole-genome sequencing (WGS).

## Introduction

Despite advances in research and development and the availability of anti-tuberculosis drugs that cure most cases of tuberculosis (TB), TB remains a major public health problem and is ranked as a leading cause of death from a single bacterial agent [1, 2].

According to the World Health Organization (WHO)'s latest global report, there were 10 million incident cases of TB in 2017 with 1.3 million deaths among HIV-negative people, and an additional 300 000 deaths among HIV-positive, of which 90% were adults. In the same year, 558 000 new TB cases with resistance to Rifampicin (RRTB) were reported, of which 82% (460 000) were multidrug-resistance to TB (MDR-TB) [1]. Ethiopia has an estimated incidence of 223 per 100000, a prevalence of 212 per 100000, and TB death of 32 per 100,000 with estimated MDR-TB rates of 1.6% and 12% among new and cases that were retreated, respectively [3, 4].

The latest national TB drug resistance survey estimates a rate of 2.3% MDR-TB among new cases, which indicates a significant increase from 1.6% from the first report [3, 5]. A major challenge is the strikingly low coverage of second-level laboratory facilities, as shown by the fact that only 1% of bacteriologically confirmed new TB cases and only 4.4% of retreatment cases had undergone drug susceptibility testing (DST) [4].

Despite a slightly decreasing number of reported drug-susceptible TB cases in the country (1), the increasing number of MDR-TB shows the need for better progress. The emergence and spread of drug-resistant TB are mainly a result of interdependent factors emanating from

the patient, the health system [6], and the bacteria's natural response under selection pressure to enter into more severe, untreatable, and undetectable forms within the host. Notably, *M. tuberculosis* survives under various stress conditions and environmental niches through its long-term dormancy, mainly modulated by its toxin-antitoxin systems [7, 8].

During the last few years, the introduction of several genotyping methods has facilitated molecular epidemiologic studies on TB [9, 10]. Large sets of databases and applications are now available for classification of clinical isolates into phylogenetic lineages at the strain level [11, 12]. These methods combined with DST are becoming valuable tools to understand the distribution and transmission dynamics of drug-susceptible, drug-resistant (DR), multi-drug-resistant (MDR), and extensively drug-resistant (XDR) TB.

Using these techniques, recent studies and reviews on molecular diversity of *M. tuberculosis* in Ethiopia extensively described *M. tuberculosis* strains from different regions [13–15]. However, they failed to address peripheral and emerging settings like the Benishangul Gumuz region, mainly because of logistics and the limited information available.

Therefore, the current study was designed to assess the drug susceptibility pattern and genetic diversity of *M. tuberculosis* strains from pulmonary TB patients in the Benishangul Gumuz and its surroundings in Ethiopia.

## Materials and methods

### The study settings

This study was conducted in the Benishangul Gumuz regional state and its surroundings, located about 700 km from Addis Ababa adjacent to the Ethio-Sudan border in the northwest of the country. The region is one of nine regional administrative states of the country. It shares borders with the Amhara region in the north and northeast, the Sudan Republic in the west, the Gambella region in the south, and Oromia region in the southeast. It has an estimated total population of 976,000 in 2014, according to the projection based on the 2007 CSA census [16]. About 30% of the population lives below the national poverty level; hence the region is characterized as emerging due to its limited infrastructure, including health service facilities [17].

Like other regions in Ethiopia, the region is subdivided into three administrative zones, namely Assosa, Metekel, and Kemashi. The Metekel zone is the largest covering 26,272 km$^2$, which is half the area of the region.

The region has two hospitals (one additional hospital was built in 2018) and 32 health centers. There is no TB culture and DST laboratories in the region during the study. Hence, the laboratory confirmation of TB in the area solely depends on sputum smear microscopy. In the current study, the two general hospitals, Assosa and Pawe, and four health centers, Felegeselam, Mambuk, Dangur, and Gilgelbeles, were included based on the availability of sputum smear microscopy and TB clinic as part of their routine TB diagnosis and treatment service. Additional three health centers were included from the Awi zone of the Amhara region, geographically proximal to the study region (Fig 1).

### Study design

A cross-sectional study was conducted from October 2013—June 2014. The study participants were newly diagnosed sputum smear-positive pulmonary TB (PTB) cases. All PTB suspects sent for sputum smear examination were asked for informed consent, and only patients with Acid-fast-bacilli (AFB) smear-positive sputum were enrolled. Before data collection, two days of theoretical and practical training was given to the personnel collecting data mainly focusing on research ethics, documentation, safety, sample collection, storage, and patient handling.

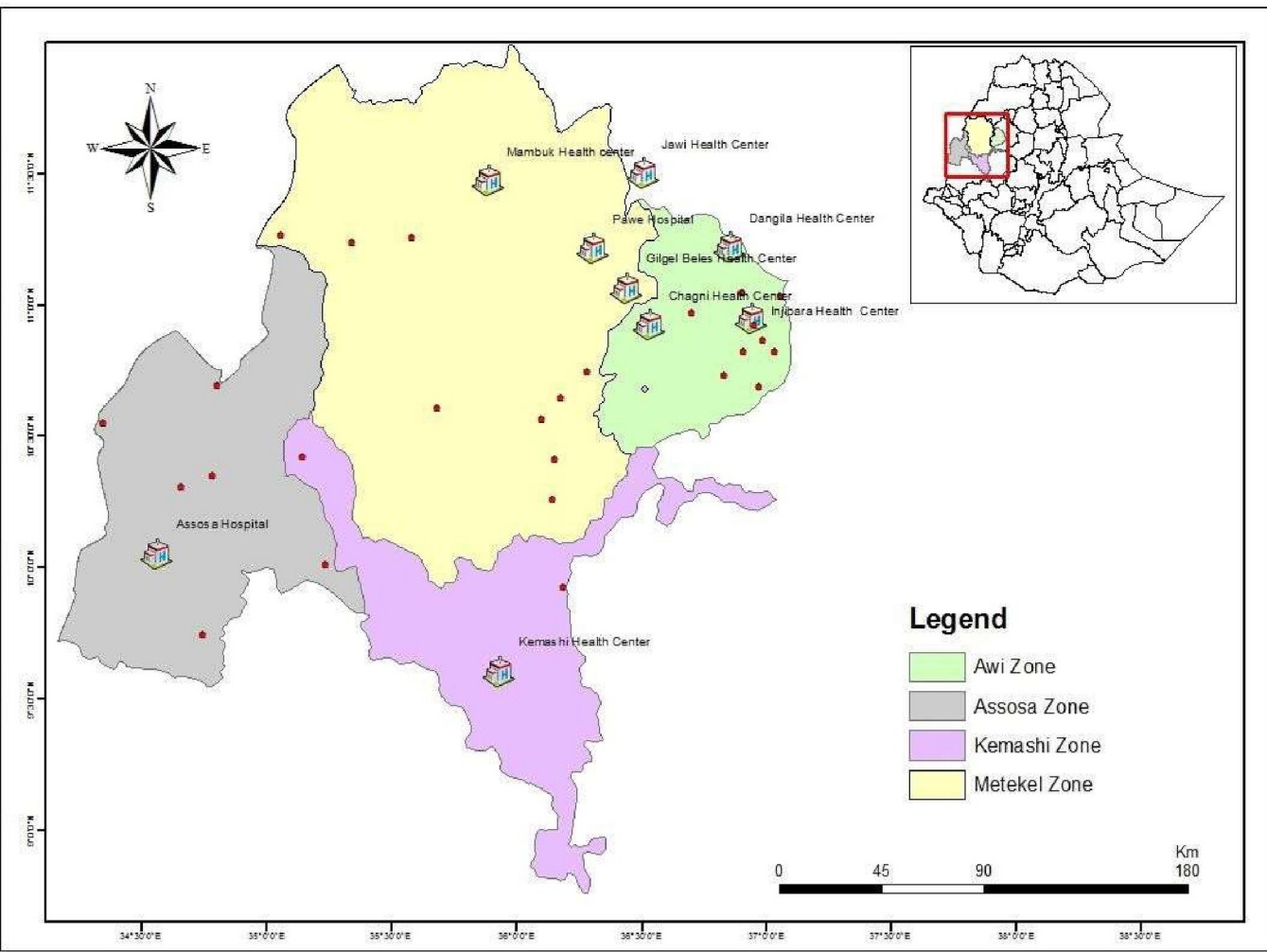

**Fig 1. Map of the study area, Benishangul Gumuz region and its surroundings.** Health institutions included in the current study are indicated by a house symbol H, and the hospitals encircled red. Dots represent towns in the region.

### Data collection

Trained nurses working at TB clinics collected sociodemographic and clinical data of participants. Trained laboratory professionals collected sputum samples. The HIV status of the participant was obtained from the patient's clinical record.

Participants were requested to give three sputum samples (Spot-Morning-Spot) as part of their routine diagnostic procedure and checked for AFB at the site of collection. AFB smear-positive sputum samples of the same patient were pooled and temporarily stored at -20°C until transported to AHRI TB laboratory at Addis Ababa, within no more than two weeks [18]. Upon arrival at AHRI, sputum samples were processed immediately by using the N-acetyl L- cysteine-sodium hydroxide (NALC-NaOH) method for sputum digestion and decontamination [19]. Fig 2 depicts an overview of the workflow for data collection and the laboratory methods applied.

### Mycobacterial culture

Egg-based *Löwenstein-Jensen* (LJ) culture medium was used following the WHO standard procedure [20]. Briefly, an equal volume of NALC-NaOH and sputum samples was left standing

**Data collection**
Sputum samples and questionnaires from TB suspects

**AFB sputum smear microscopy**
Pooled AFB smear positive sputum samples

**Sputum sample strorage**
Pooled AFB smear positive sputum samples were stored at the sites of collection at -20$^O$C for not more than two weeks

(At the health centers and hospitals)

**At study sites**

**Sputum sample transportation**
Sputum samples were transported from sites of collection to AHRI TB Laboratory while kept in ice boxes

**Sputum sample processing**
(Decontamination and digestion of sputum samples by using NALC-NaOH )

**Mycobacterial culture growth**
Löwenstein–Jensen (LJ) medium with glycerol and pyruvate

Drug susceptibility testing
(Indirect proportion method )

RD9 and Spoligotyping

**At AHRI TB Laboratory**
(Addis Ababa)

**Fig 2. Overview of the workflow for data collection and the laboratory methods.**

for 15 minutes for digestion and decontamination. Sedimentation was achieved by centrifugation at 3000 rpm for 15 minutes at +4˚C. The supernatant was discarded, and the sediment was inoculated onto two freshly prepared LJ medium slants, one containing 0.6% glycerol and the other with 0.6% sodium pyruvate to facilitate the growth of *M. tuberculosis* and *M. bovis*, respectively.

The Inoculated samples were incubated at 37˚C and followed-up weekly for eight weeks. Mycobacterial growth was provisionally confirmed by typical colony characteristics (S1 Fig) and AFB sputum smear microscopy.

## Drug susceptibility testing

DST was performed based on the indirect proportion method following standard procedures [21]. Briefly, a 24-well agar plate was prepared with 2.5 ml of Middlebrook 7H10 medium supplemented with 10% OADC and 5% glycerol per each well. The bacterial suspension was prepared in 0.2μm filter-sterilized MQ-H2O and the turbidity was adjusted to McFarland standard 1.0. Four drugs, namely Isoniazid, Rifampicin, Streptomycin, and Ethambutol, were used with critical concentration breakpoints of 0.2μg/ml, 1μg/ml, 2μg/ml, and 5μg/ml, respectively. Two drug-free wells, one with 1: 100 dilution and another containing undiluted bacteria suspension, were included as controls.

A strain was considered as "Susceptible" if no growth or considerably less than 1% growth was observed on the well containing the critical concentration of the corresponding drug compared to the control with 1% inoculum. A strain was considered as "Resistant" when growth on the well with the drug was higher than the control with 1% inoculum [22]. Any contaminated or borderline results were repeated.

## Molecular typing using RD9 and spoligotyping

Mycobacterial species and *M. tuberculosis* strain identification were carried out at AHRI TB molecular laboratory.

For molecular typing, DNA was obtained by taking a loop full colony from LJ growth and suspended in 50μl of 0.2μm filter-sterilized 1% TE buffer in 2ml tubes. The suspension was heated at 80°C in the water bath for 45 minutes followed by sonication for an additional 15 minutes. The heat-killed product was centrifuged to 13000 rpm for 10 minutes at +4°C, and the supernatant was collected in a new tube.

*M. tuberculosis* was differentiated from other species of mycobacteria based on PCR targeting RD9, as previously described by Parsons *et al*. [23]. A multiplex PCR that was designed to amplify and detect RD9 using the following primers was employed; Forward RD9_FlankFW: 5′-AACACGGTCACGTTGTCGTG-3′, Reverse RD9_FlankRev: 5′-CAAACCAGCAGCTG TCGTTG-3′, and Internal RD9_InternalR: 5′-TTGCTTCCCCGGTTCGTCTG-3′. A PCR product of 396 bp indicates the presence of RD9 which is unique to *M. tuberculosis* and is absent from all other MTB complex members. Thus, selective amplification of the region targeted by the internal primer demonstrates *M. tuberculosis*. The strains were further typed by spoligotyping based on polymorphism in the direct repeat (DR) locus using a method described previously [10, 24]. Briefly, the DR region was amplified using primers, DRa (biotin-labeled): 5′-GGTTTTGGGTCTGACGAC-3′ and DRb: 5′-CCGAGAGGGGACGGAAAC-3′. After PCR amplification, the PCR products were hybridized to 43 spacer oligonucleotides of the corresponding DR region. The hybridization patterns obtained from the reaction were converted into binary and octal formats. The mycobacterial isolates identified were assigned into Spoligotype International Types (SIT), family, and their lineages using databases [12].

## Quality control

The control strains *M. tuberculosis* H37Rv (ATCC 27294) and *M. bovis* (AF 61/2122/97) with known genotype and drug resistance patterns were run together during all laboratory assays, including growth, detection, DST, RD9 typing, and spoligotyping.

## Data analysis

All sociodemographic and clinical data of the study participants and the laboratory test results were entered and analyzed using SPSS v.24 (IBM Corp., Armonk, N.Y., USA). Association

between drug resistance, strain type, and sociodemographic and clinical data was assessed. Statistical significance was considered at a *P*-value of less than 0.05.

### Ethical statement

The research ethics review committee of Addis Ababa University and the AHRI/ALERT ethics review committee evaluated and approved the proposal (Reg. No. PO12/13). Institutional support letters were obtained from regional, zonal, and local health administrations. Study participants were provided with adequate information, and the study participants were recruited only upon signing the informed consent. The results from DST were communicated to the relevant attending physician as soon as available, for better and timely management of patients. MDR-TB cases identified were referred either to the MDR-TB treatment center at St. Peter's TB Specialized Hospital, Addis Ababa, or to Gondar University Hospital MDR-TB facility.

## Results

### Sociodemographic and clinical data of culture positive TB patients

From 107 smear-positive sputum samples collected, 81.3% (87/107) were culture positive and available for further analysis. Spoligotyping was not carried out for 12 isolates due to a technical problem, and therefore, only 75 were available for strain typing and cluster analysis.

The mean age of participants was 29 ± 9.2 SD years and ranged from 15 to 65. Participants between 18 and 47 years old accounted for 88.5% (77/87). Data on HIV status was available for 71.3% (62/87), among which 16.1% (10/62) were seropositive. There were more male patients than females with a 3:1 male to female ratio, and 55.2% of participants were married (Table 1).

### Anti-TB drug resistance pattern

Among the 87 *M. tuberculosis* isolates that were tested against Isoniazid, Rifampicin, Ethambutol, and Streptomycin (Table 2A and Fig 3A), resistance to any of the drugs tested was observed in 16.1% (14/87) of the strains.

Multidrug-resistant TB was detected in 2.3% (2/87), one of the strains was resistant to all four of the drugs tested (Table 2A and Fig 3A). The proportion of monoresistance was 6.9%, 2.3%, and 1.15% for Isoniazid, Ethambutol, and Streptomycin, respectively. There was no monoresistance to Rifampicin. Resistance to two drugs (Isoniazid and Ethambutol) in 3.45% and Polydrug resistance (resistance to more than two drugs) was observed in 2.3% of isolates. In contrast, one triple-drug resistance was identified against the Isoniazid, Ethambutol, and Rifampicin combination.

Stratifying the resistance profile into subgroups containing each drug tested showed any drug resistance of 12.6%, 8.0%, 2.30%, and 2.30% related to Isoniazid, Ethambutol, Rifampicin, and Streptomycin respectively (Table 2B and Fig 3B).

### Molecular typing and genetic diversity

PCR based RD9 typing revealed that all the 75 isolates spoligotyped were *M. tuberculosis* (S2 Fig). The spoligotyping patterns were compared to the existing strains from databases [11, 12], and assigned into octal, SIT, family, and lineages (Fig 4).

Thirty-nine distinct spoligotype patterns were identified, with 61.3% (46/75) grouped in 10 similar spoligotyping patterns containing two to 11 isolates while 38.7% (29/75) represented as unique patterns. Lineage classification revealed 72.0% (54/75), 25.3% (19/75), 1.3% (1/75) and 1.3% (1/75) isolates into Euro American, EA (Lineage 4), East African Indian, EAL (Lineage 3), East Asian, EAS (Lineage 2) and Indo-Oceanic, IO (Lineage 1), respectively (S1A Table and

**Table 1. Sociodemographic and clinical characteristics of culture positive TB patients (n = 87) from the Benishangul Gumuz region and its surroundings, northwest Ethiopia from October 2013-June 2014.**

| Characteristics | Categories | Variables | Frequency N (%) |
|---|---|---|---|
| Sociodemographic characteristics | Gender | Male | 65 (74.7) |
| | | Female | 22 (25.3) |
| | Residence | Urban | 22 (25.3) |
| | | Rural | 65 (74.7) |
| | Occupation | Farmer | 40 (46) |
| | | Merchant | 6 (6.9) |
| | | Gov't employee | 11 (12.6) |
| | | Student | 8 (9.2) |
| | | Daily laborer | 21 (24.1) |
| | | Prisoner | 1 (1.1) |
| | Educational status | Illiterate | 34 (39.1) |
| | | Grade1-4 | 10 (11.5) |
| | | Grade 5–8 | 18 (20.7) |
| | | Grade 9–12 | 16 (18.4) |
| | | Diploma and+ | 9 (6.9) |
| | Marital status | Married | 48 (55.2) |
| | | Single | 36 (41.4) |
| | | Divorced | 1 (1.1) |
| | | Widowed | 2 (2.3) |
| | Age groups | <18 | 5 (5.7) |
| | | 18–27 | 38 (43.7) |
| | | 28–37 | 31 (35.6) |
| | | 38–47 | 8 (9.2) |
| | | 48–57 | 4 (4.6) |
| | | >57 | 1 (1.1) |
| Clinical characteristics | HIV Status | Positive | 10 (11.5) |
| | | Negative | 52 (59.8) |
| | | Unknown | 25 (28.7) |
| | TB patient contact history | Yes | 20 (23) |
| | | No | 51 (58.6) |
| | | Unknown | 16 (18.4) |
| | Previous anti-TB treatment | Yes | 12 (13.8) |
| | | No | 73 (83.9) |
| | | Unknown | 2 (2.3) |

Fig 4). Three families, namely T, CAS, and H, were predominant, accounting for 61.3% (46/75) of the isolates (S1B Table and Fig 4). Among 61.3% (46/75) of the isolates containing a cluster of two to 11 similar spoligotype patterns, 71.7% (33/46) belonged to EA and 28.3% (13/46) to EAI lineages, while 17% (8/46) were without shared international spoligotype (SIT) so far and thus labeled ''New''. Thirty-eight percent (29/75) exhibited a single and unique spoligotyping pattern, with fifteen patterns having defined SIT and the remaining fourteen without SIT assigned as "New".

Notably, SIT289 (11 isolates, CAS) and SIT53 (nine isolates, T1) were the most frequent strains accounting for 14.7% (11/75) and 12% (9/75), respectively. The findings were checked for the statistical associations. None of the sociodemographic and clinical variables studied

**Table 2. A. Drug resistance pattern of first-line anti-TB drugs among culture positive TB patients (n = 87) in the Benishangul Gumuz region and its surroundings, Northwest Ethiopia, 2013/14.**

A

| Resistance profile | Resistant N (%) | Susceptible N (%) | Remark |
|---|---|---|---|
| INH | 6 (6.90) | 81 (93.10) | Monodrug resistant |
| ETB | 2 (2.30) | 85 (97.70) | |
| STM | 1 (1.15) | 86 (98.85) | |
| INH and ETB | 3 (3.45) | 84 (96.55) | Poly drug-resistant |
| INH, ETB, and RIF | 1 (1.15) | 86 (98.85) | MDR*/Poly drug-resistant |
| INH, ETB, RIF, and STM | 1 (1.15) | 86 (98.85) | |
| **Total** | **14 (16.10)** | **73 (83.90)** | |

B

| Index drug* | Categories | Resistant N (%) | Susceptible N (%) |
|---|---|---|---|
| **INH** | INH | 6 (6.90) | 81 (93.10) |
| | INH and ETB | 3 (3.40) | 84 (96.60) |
| | INH, ETB, and RIF | 1 (1.15) | 86 (98.85) |
| | INH, ETB, RIF, and STM | 1 (1.15) | 86 (98.85) |
| | Total | 11 (12.60) | 76 (87.40) |
| **ETB** | ETB | 2 (2.30) | 85 (97.70) |
| | INH and ETB | 3 (3.40) | 84 (96.60) |
| | INH, ETB, and RIF | 1 (1.15) | 86 (98.85) |
| | INH, ETB, RIF, and STM | 1 (1.15) | 86 (98.85) |
| | Total | 7 (8.0) | 80 (92.0) |
| **RIF** | INH, ETB, and RIF | 1 (1.15) | 86 (98.85) |
| | INH, ETB, RIF, and STM | 1 (1.15) | 86 (98.85) |
| | Total | 2 (2.30) | 85 (97.70) |
| **STM** | STM | 1 (1.15) | 86 (98.85) |
| | INH, ETB, RIF, and STM | 1 (1.15) | 86 (98.85) |
| | Total | 2 (2.30) | 85 (97.70) |

INH: Isoniazid, ETB: Ethambutol, RIF: Rifampicin, STM: Streptomycin; *MDR = Multidrug resistant

* "Index drug" stands for the primary drug for which grouping the resistance profile is based.

were found to be significantly associated with either drug resistance patterns or circulating strains isolated (Table 3).

## Discussion

The main aim of this study was to understand the molecular epidemiology of *M. tuberculosis* strains circulating in a remote and hard-to-reach region in order to infer lessons for interventions in these and similar areas for which no such information exists. Benishangul Gumuz region and its surroundings are among the areas in Ethiopia where resources and infrastructures are severely limited. Sociodemographic and clinical information of the study participants

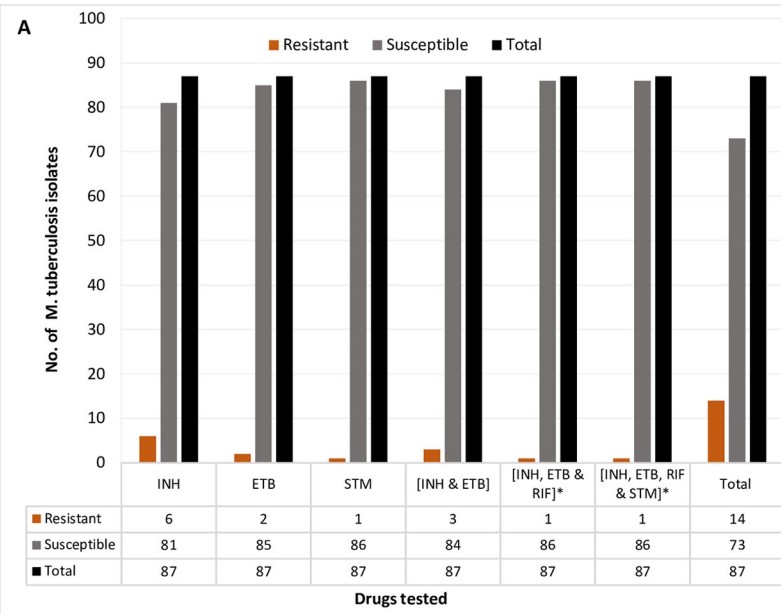

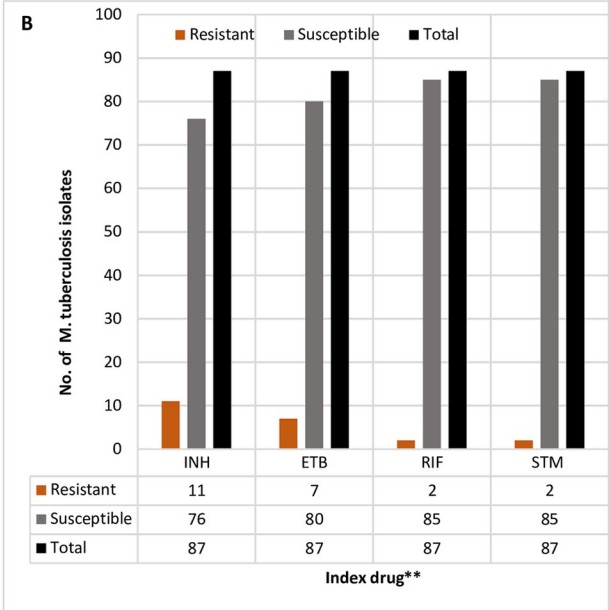

*MDR = Multidrug resistant

** A primary drug for which grouping the resistance profile is based on

**Fig 3.** A and B. The magnitude of first-line anti-TB drug resistance profile of the *M. tuberculosis* isolates from culture positive TB patients (n = 87) in the Benishangul Gumuz region and its surroundings, North West Ethiopia, 2013/14.

and the type of MTB strains isolated and their drug susceptibility patterns were analysed. Here, we showed results that either substantiated previously estimated extrapolations from other regions through new evidence to support interventions or provided baseline data for further rigorous studies in the area.

All isolates identified were confirmed as *M. tuberculosis* by PCR based RD9 typing. This finding concurs with previous reports that have shown *M. tuberculosis* as responsible for more than 90 percent of pulmonary tuberculosis infections in Ethiopia [25].

Spoligotyping revealed that 68% (51/75) of the strains belonged to groups with assigned SIT in the latest database, SITIVIT2 [12]. In comparison to the studies conducted by other groups [26–28], the remaining 32% (24/75) accounted for a slightly higher proportion of "New" strains from the area, indicating the absence of data from prior studies in the region. Our findings support the expected variation in strain distribution across different ecological niches, and the previous preliminarily results from the region [29, 30].

The identified thirty-nine distinct spoligotype patterns constituting 61.3% (46/75) of isolates clustering in two to 11 similar patterns indicate a high level of *M. tuberculosis* strain diversity in the area. The predominance of clusters within SIT289 (11 isolates) and SIT53 (nine isolates) strongly suggests an ongoing active transmission of tuberculosis in the area. Remarkably, this is despite the dispersed settlement of the population in the region.

The SIT289 strain was previously reported as predominant in Gambella and Bahir Dar. Both regions share long borders with the current study area in the southwest and northwest parts of the country, respectively [31,32]. SIT53 is frequent in central (Ambo), north (Bahir Dar, Dessie), and south Ethiopia (Dilla) [32,33]. This study site has, despite its relative isolation, an increasingly mixed population, including people originating from different parts of the country resettled in the 1980s [34], which could contribute to the diversity in the pool of circulating mycobacterial strains.

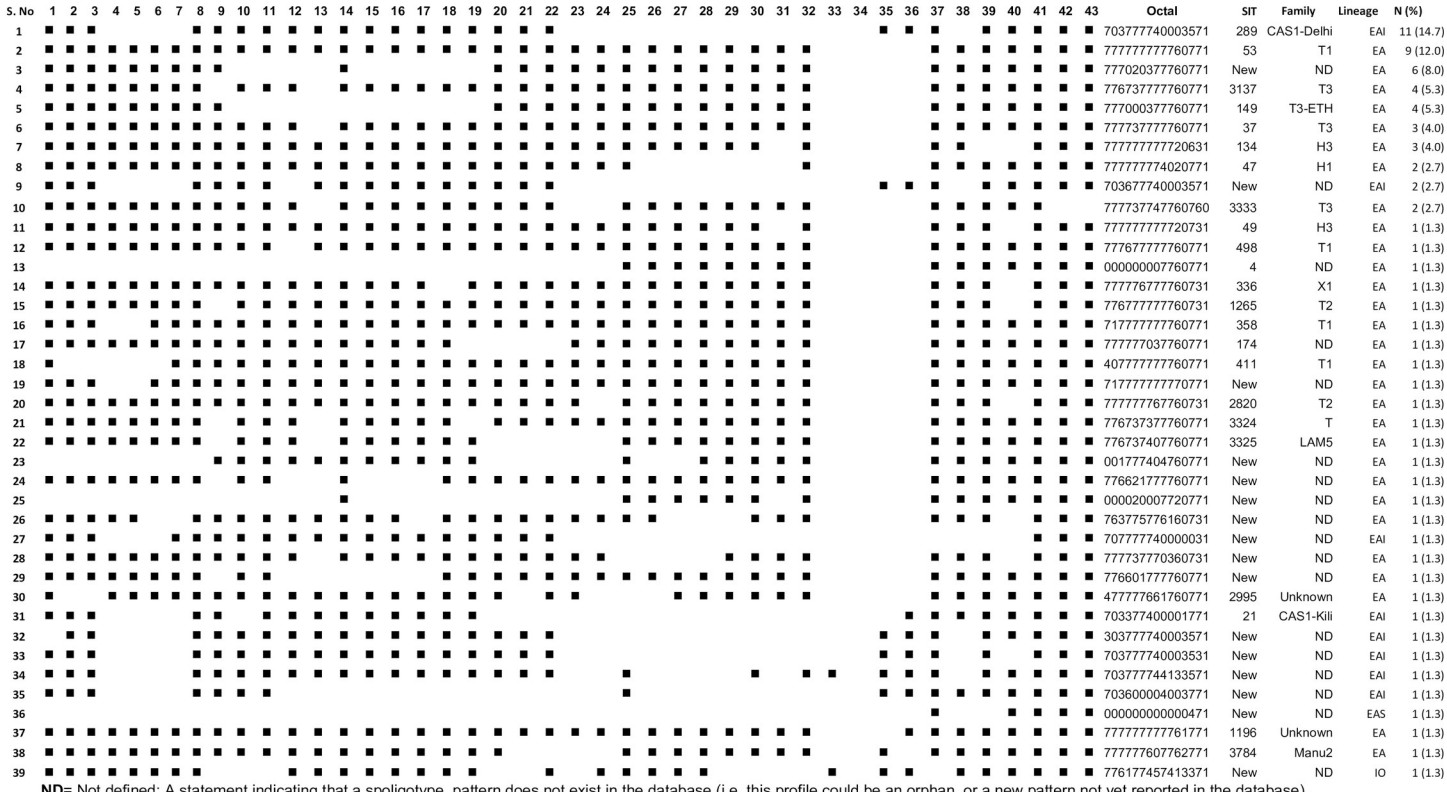

**ND**= Not defined; A statement indicating that a spoligotype pattern does not exist in the database (i.e. this profile could be an orphan, or a new pattern not yet reported in the database).
**EA**= Euro-American, **EAI**= East-African-Indian, **EAS**= East-Asian, **IO**= Indo-Oceanic
**Unknown** = Strains with SIT number but not further classified into family

**Fig 4. Spoligotyping pattern of *M. tuberculosis* isolates from culture positive TB patients (n = 75) in the Benishangul Gumuz region and its surroundings, North West Ethiopia, 2013–14.**

The identified "New" strain of lineage 2(EAS) deserves particular attention and needs to be characterized further. Lineage 2 is, in general, still very rare in Ethiopia [35]. The Beijing family has previously been reported to be prone to drug resistance. Strains within this lineage are reported to rapidly adapt to environments with variable stress conditions, including antibiotics [36]. With increased trade and travel between East Asia and Ethiopia, the risk of multiple introductions of Lineage 2 into the country remains high. It is essential to pay special attention to this risk, monitor detected strains, and conduct further studies with broader context to better design targeted control efforts.

Strains without an assigned spoligotype, so-called "New", made up 29.3% (22/75) of all isolates, and eight strains existed in clusters of two and six isolates each. The epidemiological significance of these strains needs further investigation using clinical and molecular tools with better discriminatory power.

Lineage 3(CAS-Delhi, the principal family, or CAS) with 11 strains in a cluster, which has also been reported as the dominant mycobacterial strain circulating in Sudan [37], could hint cross-border transmission since people from the current study area share market places and enjoy free cross-border movement into Sudan [38].

The dominant prevalence of lineage 4 (EA) and lineage 3 (EAI) has already been shown in previous studies from Ethiopia [13–15, 25]. Lineage 1(IO, ancestral) seems to be more prevalent in southern regions of Ethiopia and may be shared with the Eastern African coast and Kenya [39–41]. From the same setting, in a preliminarily study from spoligotyping of 33

**Table 3. Bivariate logistic analysis of factors associated with resistance to any one of anti-TB drug.**

| Variable | | Any drug resistance | | COR (95%CI) | p-value |
|---|---|---|---|---|---|
| | | Yes N(%*) | No N(%*) | | |
| Gender | M | 8 (12.3) | 57 (87.7) | 1 | |
| | F | 6 (28.3) | 16 (72.7) | 0.37 (0.11,1.24) | 0.107 |
| Residence | Urban | 1 (4.50) | 21 (95.5) | 1 | |
| | Rural | 13 (20.0) | 52 (80.0) | 0.19 (0.02, 1.55) | 0.121 |
| HIV status | Positive | 1 (15.4) | 9 (84.6) | 1 | |
| | Negative | 8 (10.0) | 44 (90.0) | 0.73 (0.21,2.58) | 0.614 |
| | Unknown | 5 (20.0) | 20 (80.0) | 0.44 (0.45, 4.37) | 0.487 |
| Previous treatment history | Yes | 5 (31.3) | 11 (68.8) | 1 | |
| | No | 9 (12.7) | 62 (87.3) | 3.13 (0.88,11.12) | 0.078 |
| Previous contact with TB patient | Yes | 6 (22.2) | 21 (77.8) | 1 | |
| | No | 8 (13.3) | 52 (86.7) | 1.86 (0.58,6.0) | 0.301 |
| Education status | Illiterate | 7 (20.6) | 27(79.4) | 1 | |
| | Literate | 7 (13.2) | 46 (88.8) | 1.70 (0.54,5.38) | 0.078 |
| Lineages | Lineage 1 (IO) | 0 (0.0) | 1(100.0) | NA | NA |
| | Lineage 2 (EAS) | 0(0.0) | 1(100.0) | NA | NA |
| | Lineage 3 (EAI) | 4(26.7) | 11(73.3) | NA | NA |
| | Lineage 4 (EA) | 9(16.1) | 47(83.9) | 1.40 (0.043, 44.9) | 0.85 |
| Strains (SIT) | New | 5 (15.2) | 28 (84.8) | 1 | |
| | Previously defined | 8 (19.0) | 34 (91.0) | 0.76 (0.22,2.58) | 0.66 |

*% calculated by row

isolates, the only two lineages reported were lineage 4 (EA) and lineage 1 (IO, ancestral). Moreover, there were 63.6% (22/33) of strains with "New" spoligotype patterns that have not been previously reported [30]. This finding highlights the need for further investigation of the molecular epidemiology of *M. tuberculosis* in the vast border regions of the country to understand trends better and respond to emerging challenges.

The observed magnitude of MDR-TB and resistance to any one first-line anti-TB drug is comparable, while Isoniazid monoresistance is considerably higher than neighbouring sites. Importantly Rifampicin, Streptomycin, and Ethambutol monoresistance were lower than in most previous studies elsewhere in the country [40–44]. The reasons need to be investigated but could include factors related to the study design, study period, sample size or study population (Urban/rural, access to drugs, health-seeking behavior, mobility of population), among others.

This study has limitations such as the sample size was relatively small, and the yield of culture was further reduced because of logistic challenges. Furthermore, there are no data on drug resistance mutations among the identified *Mycobacterium* strains. We relied on spoligotyping alone to detect transmission trends, a method that tends to overestimate rates and indicates older events than whole-genome sequencing, which is increasingly accepted as a new gold standard. Nevertheless, the number and diversity of isolates detected provided adequate data with interesting findings and meaningful conclusions. Although the indications from spoligotyping may lack precision in the magnitude of transmission, the interpretation we made from the findings that there is active ongoing transmission in the area is valid [38].

In summary, the results provided insight into the molecular diversity and drug resistance pattern of TB strains circulating in a peripheral predominantly rural region of Ethiopia where the population density is low, but where scattered urban centers have highly mobile

populations and serve as hubs of cross border trade and travel. The findings show that there is much diversity but also clustering in circulating *M. tuberculosis* strains in this setting where cross border movements are frequent and where health care services are generally scanty and inadequate. The study has also revealed the presence of several yet unknown strain types whose significance in driving tuberculosis is unexplored. In addition, there are indications that multidrug resistance is emerging and a strain that belongs to lineage 2 has been detected. It is, therefore, important to build local DST facilities in the area to detect and treat MDR TB early. It is critical to monitor the molecular epidemiology of tuberculosis in the Benishangul Gumuz region and similar regions of the country closely to predict trends in transboundary transmission and respond with appropriate interventions.

## Supporting information

**S1 Fig. Colony characteristic used for identification of the growth of *Mycobacterium tuberculosis*.**
(PDF)

**S2 Fig. PCR for RD9 deletion typing.**
(PDF)

**S1 Table.** A and B. Frequency of *M. tuberculosis* family and Lineages.
(PDF)

## Acknowledgments

The authors would like to thank all the study participants. We would also like to extend our gratitude to Mr. Shiferaw Bekele for his assistance and guidance during laboratory work, and Mr. Legesse Negash during data entry and statistical analysis. We are thankful to medical laboratory professionals and nurses working at TB clinics of the study area for their commitment and dedicated assistance during data collection. The authors would like to mention and appreciate the Benishangul Gumuz regional health bureau and Pawe woreda TB program coordinating offices for facilitating logistics for the transportation of the identified MDR-TB patients to the treatment initiation centers (TICs). Lastly, we acknowledge Mari Kaarbo for her extensive proofreading of the manuscript.

## Author Contributions

**Conceptualization:** Tekle Airgecho Lobie, Abraham Aseffa.

**Data curation:** Tekle Airgecho Lobie.

**Formal analysis:** Tekle Airgecho Lobie.

**Methodology:** Tekle Airgecho Lobie.

**Project administration:** Abraham Aseffa.

**Supervision:** Yimtubezinash Woldeamanuel, Daniel Asrat, Demissew Beyene, Magnar Bjørås, Abraham Aseffa.

**Writing – original draft:** Tekle Airgecho Lobie.

**Writing – review & editing:** Tekle Airgecho Lobie, Yimtubezinash Woldeamanuel, Daniel Asrat, Demissew Beyene, Magnar Bjørås, Abraham Aseffa.

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
