## [Decision Letter · Decision Letter 0]

10 Jan 2020

PONE-D-19-25370

Genetic Diversity and Drug Resistance Pattern of Mycobacterium tuberculosis Strains from Pulmonary Tuberculosis Patients in Benishangul Gumuz Region and Its Surroundings, Northwest Ethiopia

PLOS ONE

Dear Mr. Lobie,

Thank you for submitting your manuscript to PLOS ONE. After careful consideration, we feel that it has merit but does not fully meet PLOS ONE’s publication criteria as it currently stands. Therefore, we invite you to submit a revised version of the manuscript that addresses ALL points raised during the review process, in particular, an interest of this study should be better demonstrated and it should be better placed in the context of the published papers on Ethiopia.

English language should be edited by a native speaker (and expert in the field).

We would appreciate receiving your revised manuscript by Feb 24 2020 11:59PM. To enhance the reproducibility of your results, we recommend that if applicable you deposit your laboratory protocols in protocols.io, where a protocol can be assigned its own identifier (DOI) such that it can be cited independently in the future. For instructions see: http://journals.plos.org/plosone/s/submission-guidelines#loc-laboratory-protocols

We look forward to receiving your revised manuscript.

Kind regards,

Igor Mokrousov, Ph.D., D.Sc.

Academic Editor

PLOS ONE

Journal Requirements:

'...we would like to mention and appreciate that financial and logistic support for the transportation of the identified MDR-TB patients was facilitated by Benishangul Gumuz regional health bureau and Pawe woreda TB program coordinating office.'

'The funders had no role in study design, data collection and analysis, decision to publish, or preparation of the manuscript.'

Please provide an amended Funding Statement that declares *all* the funding or sources of support received during this specific study (whether external or internal to your organization) as detailed online in our guide for authors at http://journals.plos.org/plosone/s/submit-nowPlease state what role the funders took in the study.  If any authors received a salary from any of your funders, please state which authors and which funder. If the funders had no role, please state: "The funders had no role in study design, data collection and analysis, decision to publish, or preparation of the manuscript."

3. Please upload a new copy of Figure 1 as the detail is not clear. Please follow the link for more information: http://blogs.PLOS.org/everyone/2011/05/10/how-to-check-your-manuscript-image-quality-in-editorial-manager/

4. Please include a caption for figure 3.

5. Your ethics statement must appear in the Methods section of your manuscript. If your ethics statement is written in any section besides the Methods, please move it to the Methods section and delete it from any other section. Please also ensure that your ethics statement is included in your manuscript, as the ethics section of your online submission will not be published alongside your manuscript.

Reviewers' comments:

Reviewer's Responses to Questions

**Comments to the Author**

1. Is the manuscript technically sound, and do the data support the conclusions?

Reviewer #1: Yes

Reviewer #2: Partly

2. Has the statistical analysis been performed appropriately and rigorously? 

Reviewer #1: N/A

Reviewer #2: Yes

3. Have the authors made all data underlying the findings in their manuscript fully available?

Reviewer #1: Yes

Reviewer #2: Yes

4. Is the manuscript presented in an intelligible fashion and written in standard English?

Reviewer #1: Yes

Reviewer #2: No

5. Review Comments to the Author

Reviewer #1: I found several works on the genotyping of MTB in Benishangul Gumuz Region and in Ethiopia.

(Disassa H., A Preliminary Study on Molecular Characterization of Mycobacterium tuberculosis in Benishangul Gumuz Region, Western Ethiopia /British Microbiology Research Journal 10(6): 1-10, 201). Perhaps the author should compare their results with the data in this publication.

Tulu B. et al. Spoligotyping based genetic diversity of Mycobacterium tuberculosis in Ethiopia: asystematic review BMC Infectious Diseases (2018) 18:140

Mekonnen D et al. Molecular epidemiology of M. tuberculosis in Ethiopia: A systematic review and meta-analysis/ Tuberculosis Volume 118, September 2019, 101858.

Perhaps you should compare your results with the data in this publication.

The following errors were identified in the article:

129 line is not clear what is AHRI, where is this organization located?

135-136 lines are not the correct interpretation, because decontamination does not occur before, but after mixing within 15 minutes, and centrifugation - not 3000 rpm, but 3000 g should be, and temperature? It must be 4 C.

148 - 149 the term minimum inhibitory concentration of a breakpoint is better replaced by critical concentrations

183 error in numbers 81.3% (75/87)

188 - 189 woman and man were mixed up

338 - 339 to talk about «on-going active transmission of tuberculosis» is not right if spoligotyping is used

Reviewer #2: The authors studied a panel of 87 MTB isolates from TB patients from NorthWest Ethiopia (Benishangul Gumuz and its surroundings). The main goal was to investigate drug susceptibility patterns and genetic diversity of MTB Strains in this setting.

The subject is scientifically important but the novelty for science provided by this paper is meaningless; the study is somehow old fashioned. Indeed, Disassa et al. (2015) conducted a preliminary study for molecular characterization of MTB strains in the same area (n=33) from November 2012 to April 2013 . Moreover, Tulu and Ameni. (2018) made a review of articles published on M. tuberculosis strains and lineages in Ethiopia (up to 21 articles with spoligotyping results). SITs from many regions of Ethiopia have been extensively described and well documented. The authors report the results of routine drug susceptibility tests to first line anti TB drugs. Although cultures and DSTs and Gold standard tests, the authors did not use any molecular method to assess drug resistant TB. For genetic diversity, authors used spoligotyping which tend to overestimate links between isolates which is inconclusive for transmission routes of TB.

Other aspects that could be addressed by authors to strength the paper are:

1) To perform 15 or 24 loci MIRU-VNTR genotyping to strengthen data on MTB diversity.

2) To determine the mutations related to resistance to main anti TB drugs i.e. INH and RIF and to compare these genotyping results with those of conventional tests (drug susceptibility testing).

3) To add data regarding TB prevalence in northwest Ethiopia and give more details for the choice of this setting.

4) To review methodology section as they are too much details of well-known procedures ( for sputa collection and media preparation)...Also, there was no brief description for PCR targeting the RD9 and for spoligotyping method.

5) The autors have to check the % of the resistance to each drug, i have noticed many discordances between tha main text and table II (i.e. Rif).

6) The paper requires extensive English editing.

6. PLOS authors have the option to publish the peer review history of their article (what does this mean?). If published, this will include your full peer review and any attached files.

Reviewer #1: No

Reviewer #2: Yes: Imane CHAOUI

---

## [Author Response · Author response to Decision Letter 0]

10 Feb 2020

Response to the journal requirements (Editor) and Reviewers

Journal Requirements (response to the editor):

As the requirement and requested by academic editor of the journal the authors adhered to templates provided and found it useful in shaping the manuscript. 

To mention title, current address, corresponding author’s initial, fonts of title and subtitles, references, figure and tables and supplementary information among other points were addressed.

'...we would like to mention and appreciate that financial and logistic support for the transportation of the identified MDR-TB patients was facilitated by Benishangul Gumuz regional health bureau and Pawe woreda TB program coordinating office.'

'The funders had no role in study design, data collection and analysis, decision to publish, or preparation of the manuscript.'

 Please provide an amended Funding Statement that declares *all* the funding or sources of support received during this specific study (whether external or internal to your organization) as detailed online in our guide for authors at http://journals.plos.org/plosone/s/submit-now

Please state what role the funders took in the study. If any authors received a salary from any of your funders, please state which authors and which funder. If the funders had no role, please state: "The funders had no role in study design, data collection and analysis, decision to publish, or preparation of the manuscript."

Funding related text from the acknowledgement section of the manuscript is removed and re-written as follow on line no. 366-368 :-

‘’… Lastly, we would like to mention and appreciate the Benishangul Gumuz regional health bureau and Pawe woreda TB program coordinating offices for facilitating logistics for the transportation of the identified MDR-TB patients to the treatment initiation centers (TICs). ‘’

Funding statement, (this will be move to the cover letter as mentioned by the academic editor)

Amendment to the funding statement declaration is included in the cover letter and stated as follow. 

‘’This work is fully supported by AHRI core budget. AHRI receives core financial support from Swedish International Development Cooperation Agency (SIDA), Norwegian Agency for Development Cooperation (NORAD), and from the Government of Ethiopia through Ministry of Health. 

‘’AA holds a grant from Human Hereditary and Health in Africa (H3Africa) [H3A/18/003] implemented by the African Academy of Sciences (AAS) and the NEPAD Agency's Alliance for Accelerating Excellence in Science in Africa (AESA) in partnership with Wellcome and a grant from the National Institutes of Health (NIH) Fogarty International Center Global Infectious Diseases entitled “Ethiopia-Emory TB Research Training Program” (D43TW009127).’’

Any statements related to funding is removed from the manuscript. 

3. Please upload a new copy of Figure 1 as the detail is not clear. Please follow the link for more information: http://blogs.PLOS.org/everyone/2011/05/10/how-to-check-your-manuscript-image-quality-in-editorial-manager/

Figure 1 is checked for its quality using PACE as recommended and clear and the corrected version is uploaded.

 4. Please include a caption for figure 3.

Figure 3 is now renamed as ‘Figure 4 ‘and caption is added as ‘’ Figure 4. Spoligotyping pattern of M. tuberculosis isolates from pulmonary TB patients from the Benishangul Gumuz region and its surroundings in North West Ethiopia, 2013/14.’’. 

 5. Your ethics statement must appear in the Methods section of your manuscript. If your ethics statement is written in any section besides the Methods, please move it to the Methods section and delete it from any other section. Please also ensure that your ethics statement is included in your manuscript, as the ethics section of your online submission will not be published alongside your manuscript.

Ethics statement is REVISED and moved to the method section (line no. 198-206). 

‘’Ethical statement

The research review committee from Addis Ababa University and AHRI/ALERT health research and ethics review committee evaluated and approved the proposal with the project Reg. No. PO12/13 (S1 File). Institutional support letters obtained from regional, zonal, and local health administrations. Study participants provided with sufficient information, and only volunteers recruited after signing informed consent. Results from DST was communicated to the relevant caregiver as soon as available, for better and timely management of patients. MDR-TB cases identified were referred to either the MDR-TB treatment center at St. Peter TB specialized hospital, Addis Ababa, or to Gondar University Hospital MDR-TB facility, Gondar.’’

Response to the reviewers

Thank you for the valuable and valid comments that helped us to improve our work.

By appreciating, the effort made and time invested to bring these important points to our attention, the authors addressed all the comments from the journal editor as well as reviewers as curiously as possible. Some points, which are relevant but failed to be addressed by the virtue of limited capacity of the authors, were either strongly recommended to be taken into account in the future work plan or mentioned as limitations.

Reviewer 1.

1. Comment to compare our results with data from the following published articles of

Disassa H. et al., 2015; Tulu B. et al., 2018; and Mekonnen D. et al, 2019. 

Our results were compared and further discussed using the recommended papers. 

Specifically, Disassa H. et al., 2015 is cited as Ref. 30, and its findings are compared and discussed in line with our findings as shown on lines no. 326-331,

- Findings from the systematic review papers by Tulu B. et al., 2018 and Mekonnen D. et al., 2019 were also cited as Ref. no 13 and 15, respectively; and their findings were compared, and discussed in the document lines number 92-95 and 324-325. 

- In both review papers, the results presented regarding the Benishangul Gumuz region (the current study area) were from a single preliminary study by Disassa H. et al., 2015, and the conclusions and recommendations ask for more research in settings like Benishangul Gumuz region, refer to the conclusion from Tulu B. et al, 2018 on page 8 of 10.

2. 129 line is not clear what is AHRI, where is this organization located?

- AHRI is defined as Armauer Hansen Research Institute and located at Addis Ababa, Ethiopia. Additional explanation on the funding sources of the institute is also presented and submitted on a separate letter of the funding statement to the academic editor as follow:-

’’This work is fully supported by AHRI core budget. AHRI receives core financial support from Swedish International Development Cooperation Agency (SIDA), Norwegian Agency for Development Cooperation (NORAD), and from the Government of Ethiopia through Ministry of Health’’

3. 135-136 lines are not the correct interpretation, because decontamination does not occur before, but after mixing within 15 minutes, and centrifugation - not 3000 rpm, but 3000 g should be, and temperature? It must be 4 C. 

This statement is corrected and presented on line 144-146 as follow:-

 ‘’…… Briefly, an equal volume of NALC-NaOH and sputum samples was left standing for 15 minutes for digestion and decontamination. Sedimentation was achieved by centrifugation at 3000 rpm for 15 minutes at +4OC. ‘’

4. 148 - 149 the term minimum inhibitory concentration of a breakpoint is better replaced by critical concentrations

The phrase ‘’minimum inhibitory concentrations’’ is replaced by ‘’critical concentrations’’, and indicated on line 156-158 as follow

‘’…. Four drugs, namely Isoniazid, Rifampicin, Streptomycin, and Ethambutol, were used with critical concentration breakpoints of 0.2μg/ml, 1μg/ml, 2μg/ml, and 5μg/ml, respectively.’’

It is also corrected on line 161

5. 183 error in numbers 81.3% (75/87) 

This is now corrected and stated on line 214 as ‘’…… 81.3% (87/107) were culture positive and available for further analysis’’. Calculations for % and number were also checked across the whole manuscript and corrected when find. 

6. 188 - 189 woman and man were mixed up 

The statement mentioned as ‘’There were more female patients than male with 3:1 male to female ratio’’ was wrong and replaced by the statement ‘’…. There were more male patients than females with a 3:1 male to female ratio’’ on line no. 219-220.

7. 338 - 339 to talk about «on-going active transmission of tuberculosis» is not right if spoligotyping is used. 

For the discussions made based on the finding from spoligotyping, revision has been made and conclusive presentations were corrected, as results from the spoligotyping are valid and suggestive but not conclusive. 

Line 300-302 ‘’…The predominance of clusters within SIT289 (11 isolates) and SIT53 (nine isolates) could strongly suggest an on-going active transmission of tuberculosis in the area’’, and 

line 52-53 ‘’….Clustered spoligotype patterns suggest on-going active tuberculosis transmission in the area.’’

Reviewer 2.

The authors report the results of routine drug susceptibility tests to first line anti TB drugs. Although cultures and DSTs and Gold standard tests, the authors did not use any molecular method to assess drug resistant TB. For genetic diversity, authors used spoligotyping, which tend to overestimate links between isolates that is inconclusive for transmission routes of TB. 

Other aspects that could be addressed by authors to strength the paper are:

1) To perform 15 or 24 loci MIRU-VNTR genotyping to strengthen data on MTB diversity.

2) To determine the mutations related to resistance to main anti TB drugs i.e. INH and RIF and to compare these genotyping results with those of conventional tests (drug susceptibility testing).

The authors strongly agree the need and importance of 15 or 24 loci MIRU-VNTR for genetic typing, and determining genetic mutation to the main drugs tested in study. Nevertheless, we could not able to address this only because technical, logistic and financial issues during the study period. 

Some of the reasons and pertinent justifications of our findings are stated below. 

a. There was no MIRU-VNTR in Ethiopia during the study period and still there no this facility available. Obviously, this demand to establish collaboration from abroad that may incur us many things and further delay, if it ends successfully. 

b. Though this study at the beginning was designed to address only the genetic diversity of M. tuberculosis from the Benishangul Gumuz region (where there was no data available then), we decided to add the drug susceptibility profile study of the isolates using indirect proportion method, which is recommended by WHO as a gold standard method. Unlike most studies focusing on either molecular epidemiology or drug resistance alone, our work addressed both issues that strengthened the findings, especially to the current study setting. Despite its importance to complement the observed DST results, our results on drug resistance lack molecular methods to determine which resistance gene the bacteria carry, and this is explained as a limitation ….[line 345-347]

c. Among limitations mentioned and recommendations made to be addressed in the future work for the current settings, molecular epidemiology using molecular tools with high discriminatory power like MIRU-VNTR and WGS were given emphasis. 

d. Furthermore, two systematic reviews by Tulu B. et al., 2018 (ref. 13), and Mekonnen D. et al, 2019 (Ref. 15) which were recently published and extensively addressing the big picture of genetic diversity of M. tuberculosis in the country, were mainly based on spoligotyping. Both the reviews used overlapping studies with only few (7/31) studies included by Mekonnen D. et al. addressed additional results from MIRU-VNTR/SNP and both failed to compare results from Benishangul Gumuz region because of limited results presented from a single preliminary study by Disassa H. et al., 2015. The authors of these studies were opted to conclude by recommending the need for further study from the region (the current study area). See conclusion by Tulu B. et al., 2018 on page 8 of 10. 

We therefore, strongly believe making data avail from spoligotyping with interesting findings and suggestive conclusion is valid (Ref. 10 and 38). Moreover, the drug resistance profile of the isolates will influence the need for further interventions and research in the area especially when and where there is no enough information. 

3) To add data regarding TB prevalence in northwest Ethiopia and give more details for the choice of this setting.

Prevalence of Tuberculosis in Northwest Ethiopia 

Except the national TB prevalence survey from 2011 (Ref. 3), there no regional TB prevalence for each region in Ethiopia. However, we managed to extrapolated results from individual studies done the surrounding areas to the current study region like Bahir Dar (Ref. 32), Gondar, Bahir Dar and Debremarkos (Ref. 43) West Gojjam zone (Ref.6), Dessie (Ref.34), in East Amhara (Ref. 26) and major towns in Amhara (Ref. 40) were extensively discussed in the paper. 

Findings from the review papers mentioned above and addressing molecular epidemiology of M. tuberculosis at the national level by Tulu B. et al., 2018 (ref. 13); and Mekonnen D. et al, 2019 (Ref. 15) were discussed. 

Moreover, findings from a study by Disassa H., et al., 2015 (Ref. 30) which was based on Spoligotyping of 33 mycobacterial isolates from the similar setting, were compared and discussed as stated on line no. 326-331.

 Details for choosing this setting

More details explaining the demographic and geographic parameters of the study setting were presented under ‘’study setting’’ section line no. 100-118. Briefly, the main reason to choose the current setting, as presented in lines 26-28, is ‘’Because of the supposedly high initial cost and limitation of resources, there is limited information about drug resistance patterns and circulating strains in peripheral and emerging regions of Ethiopia.’’ As mentioned in the study setting section, Benishangul Gumuz is one of the emerging regions in the country with severely limited resources in terms of infrastructure including health facilities. In addition, the recent reviews from the country are lacking information on genetic diversity of M. tuberculosis from the area and recommended the need for further studies in the emerging regions like Benishangul Gumuz region (Tulu B. et al., 2018 (ref. 13, conclusion remark on page 8 of 10); and Mekonnen D. et al., 2019 (Ref. 15)), and therefore, we believe this study helps to call further studies needed to narrow the wider gaps existing in peripheral settings including Benishangul Gumuz region.

Moreover, there no single study so far presented drug resistance profile of M. tuberculosis complemented with strain diversity in Benishangul Gumuz region, EXCEPT single preliminary study on molecular epidemiology based on spoligotyping of 33 isolates (by Disassa H. et al., 2015). 

4) To review methodology section, as they are too many details of well-known procedures (for sputa collection and media preparation)...Also, there was no brief description for PCR targeting the RD9 and for spoligotyping method.

Details from the method section are now removed, and a brief description for PCR targeting the RD9 and Spoligotying method is added. The authors briefly provided basic details of the method for the purpose of simple understanding and clarification. 

Descriptions for PCR targeting RD9 and Spoligotyping

Description for PCR targeting Spoligotyping and RD9 were explained on line 172-184 as:- 

‘’ M. tuberculosis was differentiated from other species of mycobacteria based on PCR targeting the RD9, as previously described by Parsons et al. (Ref.23). A multiplex PCR was designed to amplify and detect the RD9 using the following primers; Forward RD9_FlankFW: 5’-AACACGGTCACGTTGTCGTG-3’, Reverse RD9_FlankRev: 5’-CAAACCAGCAGCTGTCGTTG-3’, and Internal RD9_InternalR: 5’-TTGCTTCCCCGGTTCGTCTG-3’). The PCR product of 396bp was interpreted as RD9 is present indicating M. tuberculosis.

The strains were further typed by spoligotyping based on polymorphism in the direct repeat (DR) locus using a method described previously [Ref.10 and 24]. Briefly, the DR region was amplified using primers, DRa: 5’-GGTTTTGGGTCTGACGAC-3’ and DRb: 5’-CCGAGAGGGGACGGAAAC-3’. After PCR amplification, the PCR products were hybridized to 43 spacer oligonucleotides of the corresponding DR region. The hybridization patterns obtained from the reaction were converted into binary and octal formats. A database with a broad set of strains checked for SIT (Spoligotype International Type), their corresponding numbers, family, and lineages, and assigned to the isolates (Ref. 12).’’

5) The authors have to check the % of the resistance to each drug; I have noticed many discordances between the main text and table II (i.e. Rif).

To make the presentation clear and simple for the readers, we made new table (Table 2A and B) and figure (Figure 3A and 3B). The % of drug resistance to each drug and combination of drugs is now calculated and presented in elaborated fashion. 

6) The paper requires extensive English editing.

Comments on English writing were checked and addressed after having reviewed by peers and experts in the area.

---

## [Decision Letter · Decision Letter 1]

28 Feb 2020

PONE-D-19-25370R1

Genetic diversity and drug resistance pattern of Mycobacterium tuberculosis strains from pulmonary tuberculosis patients in Benishangul Gumuz region and its surroundings, Northwest Ethiopia

PLOS ONE

Dear Mr. Lobie,

Thank you for submitting your manuscript to PLOS ONE. After careful consideration, we feel that it has merit but does not fully meet PLOS ONE’s publication criteria as it currently stands. Therefore, we invite you to submit a revised version of the manuscript that addresses the points raised during the review process.

Please address additional minor comments made by the reviewer.

We would appreciate receiving your revised manuscript by Apr 13 2020 11:59PM. To enhance the reproducibility of your results, we recommend that if applicable you deposit your laboratory protocols in protocols.io, where a protocol can be assigned its own identifier (DOI) such that it can be cited independently in the future. For instructions see: http://journals.plos.org/plosone/s/submission-guidelines#loc-laboratory-protocols

We look forward to receiving your revised manuscript.

Kind regards,

Igor Mokrousov, Ph.D., D.Sc.

Academic Editor

PLOS ONE

Reviewers' comments:

Reviewer's Responses to Questions

**Comments to the Author**

1. If the authors have adequately addressed your comments raised in a previous round of review and you feel that this manuscript is now acceptable for publication, you may indicate that here to bypass the “Comments to the Author” section, enter your conflict of interest statement in the “Confidential to Editor” section, and submit your "Accept" recommendation.

Reviewer #2: All comments have been addressed

2. Is the manuscript technically sound, and do the data support the conclusions?

Reviewer #2: Yes

3. Has the statistical analysis been performed appropriately and rigorously? 

Reviewer #2: Yes

4. Have the authors made all data underlying the findings in their manuscript fully available?

Reviewer #2: Yes

5. Is the manuscript presented in an intelligible fashion and written in standard English?

Reviewer #2: No

6. Review Comments to the Author

Reviewer #2: The authors agreed on the importance of MIRU VNTR typing, they stated that this method is not established in Ethiopia because of several constraints.

The autors should keep im nind that "home" MIRU- VNTR is not as cumbersome and it does not require expansive reagents to be done.

In methodology section, the authors have to specify that the primer DRa is biotin labeled.

For PCR targeting RD9, the authors have to sepcify that the PCR product of 396bp is related to internal primers.

The authors have to reformulate the following sentence:" A database with a broad set of strains checked for SIT (Spoligotype International Type), their corresponding numbers, family, and lineages, and assigned to the isolates"

7. PLOS authors have the option to publish the peer review history of their article (what does this mean?). If published, this will include your full peer review and any attached files.

Reviewer #2: Yes: Imane CHAOUI

---

## [Author Response · Author response to Decision Letter 1]

11 Mar 2020

Response to the reviewers (a minor revision)

Thank you again for your valuable and valid comments that helped us to improve our work.

The authors find all the comments to be scientifically pertinent and, therefore, addressed as presented below. Comments are indicated by numbering from 1 to 4 and bolded.

Reviewer 2.

1. In methodology section, the authors have to specify that the primer DRa is biotin labeled.

Response: In the manuscript line no. 180 – 181: the primers used for spoligotyping are presented as:

 (….DRa (biotin-labeled):5’-GGTTTTGGGTCTGACGAC-3’and DRb: 5’-CCGAGAGGGGACGGAAAC-3’)

2. For PCR targeting RD9, the authors have to sepcify that the PCR product of 396bp is related to internal primers.

Response: On line 176-178 stated as-

‘’A PCR product of 396 bp indicates the presence of RD9 which is unique to M. tuberculosis and is absent from all other MTB complex members. Thus, selective amplification of the region targeted by the internal primer demonstrates M. tuberculosis.’’

3. The authors have to reformulate the following sentence:" A database with a broad set of strains checked for SIT (Spoligotype International Type), their corresponding numbers, family, and lineages, and assigned to the isolates"

Response: This statement is reformulated and presented on line no. 183-185 as:

‘’The mycobacterial isolates identified were assigned into Spoligotype International Types (SIT), family, and lineages using databases.’’

4. Is the manuscript presented in an intelligible fashion and written in standard English? No

Response: To improve the intelligibility and comprehensiveness of the manuscript, senior researchers with ample expertise in TB research were invited and extensively revised the manuscript; and the revised manuscript went through additional proofreading process by a native English speaker and senior scientist. This person is acknowledged as indicated on line no. 374-375.

---

## [Editor Report · Decision Letter 2]

23 Mar 2020

Genetic diversity and drug resistance pattern of Mycobacterium tuberculosis strains isolated from pulmonary tuberculosis patients in the Benishangul Gumuz region and its surroundings, Northwest Ethiopia

PONE-D-19-25370R2

Dear Dr. Lobie,

We are pleased to inform you that your manuscript has been judged scientifically suitable for publication and will be formally accepted for publication once it complies with all outstanding technical requirements.

With kind regards,

Igor Mokrousov, Ph.D., D.Sc.

Academic Editor

PLOS ONE
---

## [Editor Report · Acceptance letter]

25 Mar 2020

PONE-D-19-25370R2 

Genetic diversity and drug resistance pattern of *Mycobacterium tuberculosis* strains isolated from pulmonary tuberculosis patients in the Benishangul Gumuz region and its surroundings, Northwest Ethiopia 

Dear Dr. Lobie:

I am pleased to inform you that your manuscript has been deemed suitable for publication in PLOS ONE. Congratulations! Your manuscript is now with our production department. 

With kind regards,

on behalf of

Dr Igor Mokrousov 

Academic Editor

PLOS ONE